# A Differential Diagnosis of Unusual Gastric Ulcer

**DOI:** 10.3390/diagnostics12081929

**Published:** 2022-08-10

**Authors:** Soo-Yoon Sung, Hyun Ho Choi, Kyung Jin Seo

**Affiliations:** 1Department of Radiation Oncology, Eunpyeong St. Mary’s Hospital, College of Medicine, The Catholic University of Korea, Seoul 06591, Korea; 2Department of Internal Medicine, Uijeongbu St. Mary’s Hospital, College of Medicine, The Catholic University of Korea, Seoul 06591, Korea; 3Department of Hospital Pathology, Uijeongbu St. Mary’s Hospital, College of Medicine, The Catholic University of Korea, Seoul 06591, Korea

**Keywords:** gastric ulcer, diffuse large B cell lymphoma (DLBCL), endoscopy

## Abstract

The endoscopic findings of diffuse large B cell lymphoma have various presentations. In our case, the patient had developed multiple elevated central ulceration lesions, and the peripheral elevated portion had a heaped-up margin. The margin had a sharp, smooth edge that was not infiltrative and could be confused with a simple gastric ulcer. Endoscopists should be aware of the possibility of multiple lymphoma ulcers with heaped-up margins. We present some unusual endoscopic features of lymphoma, which are easily misdiagnosed as gastric ulcers.

A 61-year-old man visited the hospital for evaluation of persistent epigastric pain and postprandial discomfort for 4 months. He had undergone an upper endoscopy at another institution 3 months earlier and was diagnosed with a gastric ulcer. He was treated with medication, but the symptoms persisted. In our hospital, complete blood count (CBC) revealed a hemoglobin level of 6.2 g/dL, hematocrit of 21.4%, white blood cell (WBC) count of 5750 × 103/μL, and platelet count of 223 × 103/μL. The serum laboratory test results were as follows: aspartate transaminase (AST), 62 U/L; alanine transaminase (ALT), 32 U/L; alkaline phosphatase (ALP), 152 U/L; and lactate dehydrogenase (LDH), 563 U/L. Upper endoscopy revealed multiple gastric ulcers without active bleeding in the antrum. The ulcers had elevated round margins and varied in diameter from 3 to 6 mm; their base was covered with exudate (Figure 1).

A biopsy of the gastric ulcer lesion revealed dense atypical lymphoid cell infiltration with ulcerations (Figure 2a). The immunohistochemistry results were CD20-positive (Figure 2b), CD10-positive (Figure 2c), and Ki-67 of 90% (Figure 2d), consistent with DLBCL, germinal center B-cell (GCB) subtype. Further laboratory testing showed that HIV Ag/Ab was positive. HIV infection was confirmed by western blot. A positron emission tomography/computed tomography (PET-CT) scan revealed multiple lymphadenopathies on both sides of the neck, mediastinum, and abdominopelvic cavity, and lesions involving the stomach, liver, and small bowel. This patient’s final diagnosis was HIV-related diffuse large B cell lymphoma (DLBCL). Endoscopic findings of gastric DLBCL have various presentations, such as nodular, polypoid, ulcerofungating, ulceroinfiltrative, erosive, diffuse infiltrating, thickened fold-like, and mixed types [1,2,3,4]. This patient had developed multiple elevated central ulceration lesions, and the peripheral elevated portion had a heaped-up margin. The margin had a sharp, smooth edge that was not infiltrative and could be confused with a simple gastric ulcer [5,6,7]. Endoscopists should be aware of the possibility of gastric lymphoma when there are multiple ulcers with heaped-up margins.

## Figures and Tables

**Figure 1 diagnostics-12-01929-f001:**
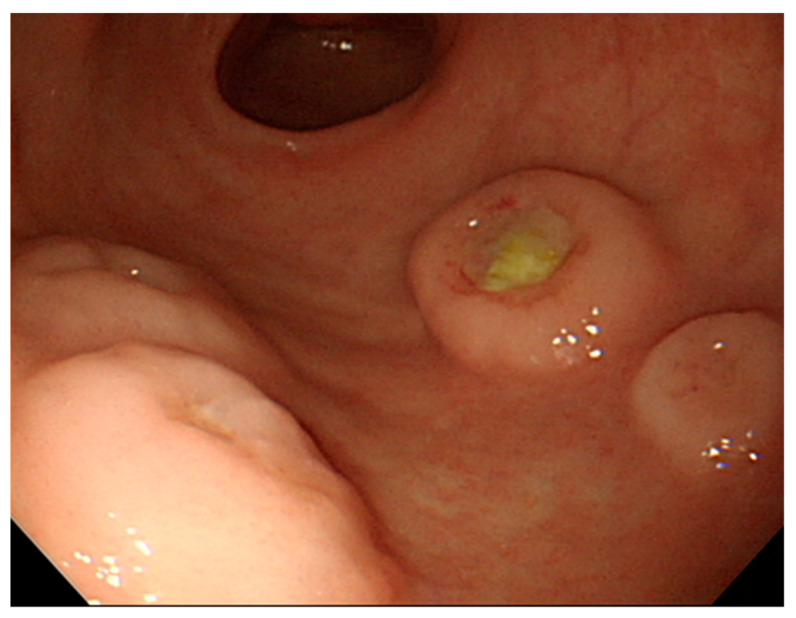
Upper endoscopy showed multiple gastric ulcers that were elevated round margin and were covered with exudate at base.

**Figure 2 diagnostics-12-01929-f002:**
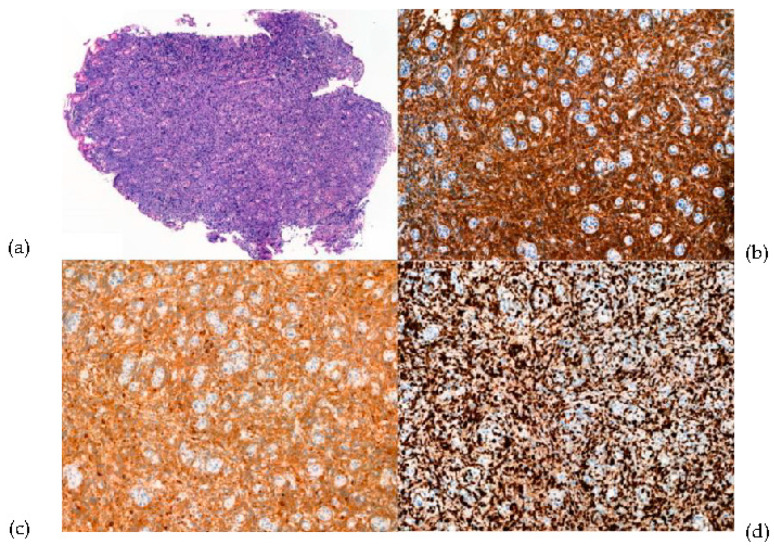
Gastric biopsy showed diffuse infiltration by atypical lymphoid cell infiltration with ulcerations ((**a**), upper left) and intense positivity for CD20 ((**b**), upper right), CD10-positive ((**c**), lower left), and Ki-67 of 90% ((**d**), lower right) at immunohistochemistry analysis.

## Data Availability

The data presented in this study are available on request from the corresponding author.

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
