# Peer review of "A Differential Diagnosis of Unusual Gastric Ulcer"

_diagnostics, 2022, doi:10.3390/diagnostics12081929_

Round 1
Reviewer 1 Report
Figure 1 is a typical endoscopic image of gastric lymphoma. Metastatic cancer from other organ is the sole differential diagnosis in this case. No endoscopy experts diagnose this image as peptic ulcer. The authors should reconsider the title as it is misleading.
Author Response
Reviewer 1
Figure 1 is a typical endoscopic image of gastric lymphoma. Metastatic cancer from other organ is the sole differential diagnosis in this case. No endoscopy experts diagnose this image as peptic ulcer. The authors should reconsider the title as it is misleading.
Response> Thank you for your sensible comments. We agree that the title may be misleading. According to your recommendation, we revised our title as follows:
A differential diagnosis of unusual gastric ulcer
Reviewer 2 Report
Please consider introducing a new paragraph after the text ref figure 2 (… - cell (GCB) subtype. Starting with: Further laboratory….
Author Response
Reviewer 2
Please consider introducing a new paragraph after the text ref figure 2 (… - cell (GCB) subtype. Starting with: Further laboratory….
Response> Thank you for your kind comments. We revised paragraph according to your advice:
The immunohistochemistry results were CD20-positive (Figue 2b), CD10-positive (Figure 2c) and Ki-67 of 90% (Figure 2d), consistent with DLBCL, germinal center B-cell (GCB) subtype.
Further laboratory testing showed that HIV Ag/Ab was positive.